# High-latitude MSTIDs over the EISCAT-3D site: Solar activity and seasonal dependency

Rahul Rathi<sup>1</sup>, Adrian Grocott<sup>1</sup>, Tim K. Yeoman<sup>2</sup>, Mark Lester<sup>2</sup>

<sup>1</sup>Lancaster University, Lancaster, United Kingdom

Correspondence: Adrian Grocott (a.grocott@lancaster.ac.uk)

Abstract. This work involves an investigation of high-latitude medium scale traveling ionospheric disturbances (MSTIDs) over the newly established EISCAT-3D radar site. We have used the ground backscatter data from an HF radar located at Hankasalmi, Finland (~62.3°N, ~26.61°E geographic coordinates), which is a part of the SuperDARN (Super Dual Auroral Radar Network). Data from solar maximum (2001 & 2014) and minimum (1996 & 2009) years from solar cycles 23 and 24 have been used to investigate the characteristics, seasonal variation, and possible generating sources of high-latitude daytime MSTIDs. Irrespective of the seasons and solar activity conditions, a dominant fraction of MSTIDs propagate equatorward with velocity in the range of 50 – 150 m s<sup>-1</sup> and period in the range of 30 – 60 minutes. Their occurrence shows seasonal and solar activity dependency. They normally occur during winter and equinoctial months. During solar maximum conditions, the occurrence was comparatively higher (~72 %) than during solar minimum years (below 50 %). Furthermore, the MSTIDs' occurrence showed a dependence on IMF Bz, being generally higher during intervals of prolonged northward or southward IMF Bz, and lower during small or fluctuating IMF Bz conditions. Our results indicate that MSTIDs occurrence showed seasonal variation as well as dependence on the solar forcing over the EISCAT-3D location. Therefore, this statistical study will help in providing comprehensive insight about the MSTIDs which will be effective in scheduling future experimental runs of EISCAT-3D to explore their 3-dimensional structures.

## 1. Introduction






Traveling ionospheric disturbances (TIDs) are propagating electron density perturbations in the ionosphere. They pose a persistent challenge due to their ability to severely affect radio propagation, often leading to disruption of radio-communications, increased convergence time of the precise point positioning of Global Navigation Satellite Systems (GNSS), and distortion of radio signals from astronomical sources (Boyde et al., 2025; Carter et al., 2023; Maletckii & Astafyeva, 2024). They are believed to be the manifestation of the atmospheric gravity waves (Hines, 1960; Hocke & Schlegel, 1996; Hunsucker, 1982). Based on their scale sizes, TIDs are categorized as large, medium, and small scale (Hunsucker, 1982). Large scale TIDs (LSTIDs) having wavelengths of more than 1000 km, propagate towards the equator with a velocity in the range of 400-1000 m s<sup>-1</sup>, have a period of more than 1 hour and are mostly generated by geomagnetic activity in the auroral region (Ding et al., 2008; Tsugawa et al., 2004). Small scale TIDs (SSTIDs) are small scale waves (wavelengths less than 100 km) and are usually generated by lower atmospheric gravity waves (Boyde et al., 2022). Medium scale TIDs (MSTIDs) normally propagate equatorward with a velocity of a few hundred m s<sup>-1</sup>, a period in range of 30 – 60 minutes, and wavelength of few hundred km (Grocott et al., 2013; Hocke & Schlegel, 1996; Huang et al., 2016; Ishida et al., 2008; Rathi et al., 2025; Shiokawa et al., 2003) and are believed to be generated by various sources (e.g., external solar forcing, internal atmospheric forcing, and natural hazards).

<sup>&</sup>lt;sup>2</sup>University of Leicester School of Physics and Astronomy, Leicester, UK








MSTIDs have been observed and reported over both mid and high-latitude regions (Grocott et al., 2013; Huang et al., 2016; Ishida et al., 2008; Shiokawa et al., 2003). Mid-latitude MSTIDs have been investigated extensively, and observations indicate that they generally occur during geomagnetic quiet time (Ding et al., 2011; Huang et al., 2016), whereas MSTIDs over high-latitude occur during both geomagnetic quiet and active times. They are believed to be generated by multiple sources such as Joule heating by geomagnetic storms/substorms, gravity waves generated by tropospheric convection, and the solar terminator (Grocott et al., 2013; Ishida et al., 2008; Prikryl et al., 2022 & 2024). There are studies which explored high-latitude MSTIDs utilizing datasets from various ground and satellite-based instruments over both hemispheres (Frissell et al., 2016; Grocott et al., 2013; Ishida et al., 2008; Negale et al., 2018; Ogawa et al., 1987; Prikryl et al., 2022 & 2024; Shiokawa et al., 2013; Vlasov et al., 2011; Xiong et al., 2025). These studies reported that MSTIDs normally propagate towards the equator and that their occurrence shows seasonal variation. However, an uncertainty has been observed in their occurrence with respect to the seasons; with peak occurrence in both summer (Vlasov et al., 2011) as well as in winter (Negale et al., 2018; Ogawa et al., 1987) months reported. In addition, there are two schools of thought on their generation with respect to geomagnetic activity. There are studies which reported high-latitude MSTIDs during geomagnetic quiet and/or moderately disturbed times and suggested that the occurrence of these MSTIDs did not increase with increasing geomagnetic activity (Frissell et al., 2016; Ishida et al., 2008; Ogawa et al., 1987). On the other hand, Prikryl et al. (2022 & 2024) reported that MSTIDs over high-latitudes can also be generated by atmospheric gravity waves induced through Joule heating by external solar forcing (geomagnetic storms and/or substorms). More recently, Xiong et al. (2025) reported that high-latitude daytime MSTIDs can also be observed during prolonged northward IMF Bz conditions and explored the possible role of intermittent lobe reconnection behind their generation. It is thus very apparent that a wide variety of generation or seeding mechanisms are associated with high-latitude MSTIDs. However, there is still much uncertainty surrounding their seasonal variation and the conditions that make a particular mechanism dominant.

This study aims to explore the seasonal variation and generating sources of the high-latitude MSTIDs using ground backscatter data of the Hankasalmi radar. The rationale behind selecting the Hankasalmi radar is the geographic location and the operational range that coincides partially with newly established EISCAT-3D radar. EISCAT-3D is the most advanced high power three-dimensional imaging radar for atmospheric, ionospheric, and near-Earth space investigations. Since it requires high power and high cost for experimental runs, a prior understanding of ionospheric irregularities (e.g., their characteristics, occurrence patterns with respect to season and external solar forcing) is required before its operational phase. In order to deepen our understanding, we have investigated daytime MSTIDs during solar maximum (2001 & 2009) and minimum years (1996 & 2009) from solar cycles 23 – 24 over this region and have used different approaches to characterize the observed MSTIDs while also investigating the role of external solar forcing behind their occurrence.

## 2. Instruments, Data, and Analyses Methods

In the present study, we have used the data from an HF radar located at Hankasalmi, Finland (~62.3°N, ~26.61°E geographic coordinates). The Hankasalmi radar (hereafter HAN) is a part of the SuperDARN (Super Dual Auroral Radar Network), an international array of coherent radars in the Northern and Southern Hemispheres (Greenwald et al., 1995; Chisham et al., 2007). In the normal mode of operation, HAN radar uses 16 beams for scanning with azimuthal separation of 3.24°, where each beam typically consists of 75 range gates of size 45 km, with the first range gate sampling at 180 km from the radar location (for a full field of view thus spanning ~3500 km). Each





beam typically has a dwell time of  $\sim$ 3 sec and thus the radar completes a full 16 beam scan in  $\sim$ 1 minute (in the initial phase of operation the radar had dwell time of  $\sim$ 7 sec with a full scan in  $\sim$ 2 minutes).

SuperDARN works on the principle of coherent scattering and receives backscatter signals from the ionosphere as well as from the ground. It receives ionospheric backscatter if the transmitted signal intersects with the ionospheric target orthogonally to the magnetic field (achieved through ionospheric refraction of the signal). Further refraction of the signals directs the transmitted signals to the ground, from which they can propagate back to the radar. This is known as ground backscatter (GBS). Modulations/fluctuations in ionospheric plasma density caused by MSTIDs affect the GBS by focussing/defocussing of the signals (Grocott et al., 2013; Samson et al., 1989 & 1990) as shown in Figure 1a. It is apparent from the figure that the location of the TID signature in GBS will be displaced from its actual location at ionospheric height. Therefore, we have mapped the GBS to the ionosphere using the standard geolocation functions in the pyDARN python library (SuperDARN Data Visualization Working Group, 2025) to estimate the location of the observed MSTIDs. The field of view (FOV) of the HAN radar in GBS ranges is represented by blue FAN plot whereas the red FAN plot represents the mapped ionospheric ranges in Figure 1b. The location of the HAN radar and EISCAT-3D at Skibotn, Norway are represented by the red and yellow dots, respectively. The EISCAT-3D is located close to beam 7 (the nearly northward pointing beam) of the HAN radar. Therefore, in this study we have used data from beam 7 (as the central beam) in our analysis. We also restrict the analysis to radar range gates 20 - 45, which is the region where the maximum amount of radar ground backscatter was observed. This region is confined between the white dashed curves (in GBS ranges) and red dashed curves (in mapped ranges) in Figure 1b.

**Figure 1.** (a) Model ray paths of HF propagation through a TID modulated ionosphere (adapted from Samson et al. (1990)). (b) Field of view (FOV) of HAN radar in GBS ranges (blue FAN plot) and ionospheric mapped ranges (red FAN plot). The red and yellow dots show the locations of the HAN radar and EISCAT-3D at Skibotn, respectively. HAN radar range gates 20 and 45 (bounding the area under study) are indicated by the white dashed (GBS ranges) and red dashed (mapped ranges) curves.

The main objective of this study is to investigate the propagation characteristics and occurrence of high-latitude MSTIDs over the EISCAT-3D location during different solar activity conditions. We have therefore used HAN GBS data for solar minimum (1996 & 2009) and maximum (2001 & 2014) years from solar cycles 23 and 24. To determine the propagation characteristics, viz. velocity, period, and direction of propagation [azimuth angle (clockwise from geographic north)] of the observed MSTIDs in the GBS, the multichannel maximum entropy








method (MULMEM) has been used (Grocott et al., 2013; Ishida et al., 2008; Shibata, 1987; Strand, 1977; Ulrych & Bishop, 1975). This method uses cross-spectral analysis to determine the parameters of MSTIDs from the GBS time series. We note that GBS depends on the ionospheric plasma density which reduces significantly during nighttime (Milan et al., 1997), therefore, the present study focusses on daytime MSTIDs observed in the GBS.

MSTIDs have periods of ~30 - 60 minutes, therefore, we have performed the MULMEM analysis on a window of 160 minutes, such that it covers at least 2 - 3 wave periods. The analysis is then performed across an 8-hour time series with the window shifted accordingly (with overlapping window as considered in Grocott et al., 2013). The MULMEM algorithm uses three time series from three radar cells (combination of beams and range gates) to determine the propagation characteristics [velocity, azimuth (clockwise from geographic north), and period] of the observed MSTIDs. The location of EISCAT-3D at Skibotn lies within the beam 7 of HAN radar, therefore, in the present study we considered beam set (5, 7, 9). The limit for range gate is based on the higher GBS occurrence and it is seen that, during daytime, consistently higher GBS was observed between range gates 20 and 45. Therefore, combinations of the beam-range sets (cells) are [(5, r); (7, r+4); (9, r)], where r (range gates) varies from 20 to 45. The MULMEM method uses the time series of three cells to detect MSTIDs and determine their parameters. Figure 2 shows examples of the parameters from different cell sets of two cases of MSTIDs. Average parameters for every half an hour window and their standard deviation (shown by error bars) are plotted for different cell sets. Figure 2a shows the temporal variation of average parameters for two different cell sets on 27 January 2014. In this case parameter values were similar for both cell sets and also did not vary with time. For such cases where parameters showed similar values for different cell sets and time, we considered those as single MSTID events. Whereas in another case (on 11 February 2014) shown in Figure 2b, parameters showed different values for different cell sets and time. For the first cell set (first row of Figure 2b), two different values of the parameters were observed (marked with black rectangles) for two different time ranges. For such cases, where parameters showed two different values (azimuth difference was close to or more-than 90°; velocity & period difference was close to or more than double) either with time and/or different cell sets, we counted them as two different MSTIDs. The number of such cases are very less and in most of these cases MSTIDs were observed at different times. The cases where the parameters showed high variation with larger standard deviations and/or scattered values with time (as shown in second and third row of Figure 2b) were not considered/counted. Further, to confirm the presence of MSTIDs and their propagation, we have also checked the Range Time (RT) (e.g. Figure 3a) and FAN (which shows spatial coverage of GBS power across the radar's FOV, see Figure S1 in supporting information) plots for each case.

There were a number of cases where MULMEM failed to identify MSTIDs in the RT plots and was unable to determine their characteristics. We presume this is due to the requirement for a clear signal to be present in the 3 radar cells. In many of these cases, a signal was present in at least one cell, enabling a Fast Fourier Transform (FFT) analysis to determine a dominant period. For all the MSTID events observed in the RT plots we therefore also performed an FFT analysis to determine their period (using the same window size from the time series of beam 7) by selecting time series between the range gates 20 and 45 (as selected for MULMEM). Figure 3 shows an example MSTID (Figure 3a), and its parameters obtained from MULMEM (red curves in Figures 3b-d: velocity, azimuth angle, & period) & FFT (blue curve in Figure 3d: period). We have also provided a comparison of the MSTIDs detected using the MULMEM analysis, and the total number of MSTIDs observed in the RT plots for each year (see Table 1).

Figure 2. Parameters (azimuth, velocity, and period) determined using MULMEM for the MSTIDs observed on two different days; (a) on 27 January 2014 and (b) on 11 February 2014.

**Figure 3.** (a) An example showing MSTID in RT plot, (b-c) velocity and azimuth angle (clockwise from geographic north) determined using MULMEM, (d) comparison of period determined using MULMEM (red curve) and FFT (blue curve).

#### 3. Results








This section presents a statistical overview of the high-latitude daytime MSTIDs over EISCAT-3D during different solar activity conditions. As described in the previous section, we have applied different methods to detect and determine their characteristics. A detailed description about their occurrence and characteristics is given below.

#### 3.1 Seasonal and solar activity dependence of MSTIDs' occurrence

As observed and reported earlier, SuperDARN GBS depends on a number of factors including the refraction and absorption of the transmitted signal, the occurrence of plasma structures, and the ionospheric density. In general, the higher the ionospheric density the greater will be the level of GBS (Milan et al., 1997). In the present study, due to higher ionospheric density in the daytime and thus higher GBS, we have considered the daytime (between 9 and 17 LT close to the beam 7 observation region) observations from the HAN radar during the selected four years. Figure 4 shows the variation of monthly average F10.7 index, percentage occurrence of GBS, and MSTIDs percentage and relative occurrence. In each subfigure, the maroon and grey curves show the variation during solar maximum (dashed for 2001 & solid for 2014) and minimum (dashed for 1996 & solid for 2009) years, respectively. Figure 4a shows the monthly variation of F10.7 index, which was comparatively higher during solar maximum years (2001 & 2014) than during minimum years (1996 & 2009). Figure 4b shows the variation of the monthly percentage occurrence of daytime GBS of beam 7. The GBS monthly percentage occurrence was determined by dividing the number of bins (time and range gate bin) where GBS was present (GBS power > 0) by total number of bins for time range 8 - 16 UT and range gate 20 - 45 for each day (when radar was operation in its normal mode) of the respective month. It is to be noted that, irrespective of the solar activity conditions, GBS occurrence across all the years showed a seasonal variation. GBS occurrence was comparatively higher in the winter and equinoctial months, but it reduced significantly in the summer. In addition to the seasonal variation, GBS occurrence showed a solar activity dependence. It was higher during the solar maximum years (maroon curves) than during the minimum years (grey curves). Figure 4c shows the monthly percentage occurrence of MSTIDs, calculated as the total number of days with MSTIDs activity observed in GBS for each month with respect to the total number of days SuperDARN was operational (in normal mode) for that particular month. The occurrence of MSTIDs showed a similar pattern to the GBS occurrence, with a significantly higher occurrence during solar maximum years, and during winter and equinoctial months, compared to solar minimum years and summer months. It can also be seen from Table 1 that the occurrence of MSTIDs was more than 70 % during solar maximum years (2001 & 2014), whereas it reduced below 50 % during the solar minimum years (1996 & 2009). Figures 4b & c shows that both GBS and MSTIDs occurrence exhibited seasonal variation. Therefore, there are chances that MSTIDs seasonal variation might have influenced by the GBS seasonal variation. We have thus determined the relative occurrence (see Figure 4d) of the MSTIDs with respect to GBS occurrence, which was determined by dividing the MSTIDs percentage occurrence for each month with the percentage occurrence of GBS for the respective month. It is interesting to note that, similar to the GBS and MSTID occurrence, MSTIDs relative occurrence also showed seasonal variation.

**Figure 4.** (a) Monthly average F10.7 index, (b) monthly % occurrence of ground backscatter, (c) monthly % occurrence of the MSTIDs, and (d) relative occurrence of MSTIDs with respect to GBS occurrence during solar minimum and maximum years of solar cycle 23 and 24.

# 185 3.2 Characteristics of MSTIDs

Figure 5 presents bar plots of the velocity and period distribution of the observed MSTIDs from the MULMEM and FFT analyses described above. The bar plots in brown & maroon show the MSTIDs' percentage occurrence (obtained by dividing the number of MSTIDs in each group of velocity/period to the total number of MSTIDs





observed in the respective year) during solar maximum years (2001 & 2014), whereas dark & light grey bars show MSTIDs during solar minimum years (1996 & 2009). Figure 5a shows the velocity distribution obtained using the MULMEM method, which is distributed in 5 velocity groups (0-50, 50-150, 150-250, 250-350, & above 350; in m s<sup>-1</sup>). It can be inferred from the velocity distribution that irrespective of the solar activity conditions (across 4 years), ~60 % of the MSTIDs propagated in the velocity range 50-150 m s<sup>-1</sup>. The second dominant fraction (more than 20 %) of MSTIDs propagated with velocity in the range of 150-250 m s<sup>-1</sup>. Figures 5b-c show the period distribution determined using both MULMEM and FFT methods. This shows that irrespective of the methods and solar activity conditions, most of the MSTIDs have period in the range of 30-60 minutes, whereas the second dominant fraction has period in the range of 60-90 minutes.

Figure 6 shows the distribution of the azimuth angle of the observed MSTIDs. The azimuth is distributed in 8 groups with each group covering an angle range of 45 degrees (centred at 45, 90, 135, 180, 225, 270, 315, & 360 degrees). Each circle in the plots represents magnitude of the velocity in an increasing order from the centre (each circle marked with the magnitude of the velocity in red font with 0 m s<sup>-1</sup> at the centre). The grey shaded areas represent the different azimuth ranges, and their extension represents the average velocity of the MSTIDs traveling in those directions. The numbers of MSTIDs traveling in each azimuth range are marked by the yellow font on each shaded area. We can infer that, regardless of the solar activity conditions, a dominant fraction of the observed MSTIDs propagated equatorward (southward) with a second dominant fraction propagating southeastward. There is a very small number of MSTIDs traveling towards north, east, and west. It is also interesting to note that the MSTIDs propagating meridionally (equatorward and/or poleward) had lesser velocity as compared to those propagating zonally (eastward and/or westward). However, there is a distinct imbalance in the number of zonally and meridionally propagating MSTIDs, therefore, comparison may provide biased results. We have thus performed the significance test (Fisher Exact test) for the obtained results, and it showed that this result is significant (with p value less than 0.0001). Table 1 also shows the dominant parameters (velocity range, period range, and azimuth angle) of the MSTIDs during all the four years. In addition, we have also assessed the characteristics with respect to seasons, however, we did not find any significant dependency on seasons observed in their occurrence (figure not provided).

Figure 5. Velocity and period distribution of the MSTIDs observed during the four years of the two solar cycles.

Figure 6. Distribution of the azimuth angle of MSTIDs, number of MSTIDs in each azimuth range (yellow values), and their average velocity (radial axis, labelled with red values).

**Table 1.** Showing the year wise MSTIDs' percentage occurrence (detected by MULMEM and total observed in the RT plots) and their dominant characteristics (velocity, period, & azimuth).

| Year<br>(solar activity) | Yearly Occurrence (%) |                        | Dominant Vel.        | Dominant     | Dominant     |
|--------------------------|-----------------------|------------------------|----------------------|--------------|--------------|
|                          | Detected by MULMEM    | Total<br>(in RT plots) | (m s <sup>-1</sup> ) | Period (min) | Prop. Direc. |
| 1996 (min)               | ~ 19                  | ~ 41                   | 50 – 150             | 30 – 60      | Equatorward  |
| 2009 (min)               | ~ 25                  | ~ 50                   | 50 – 150             | 30 – 60      | Equatorward  |
| 2001 (max)               | ~ 46                  | ~ 72                   | 50 – 150             | 30 – 60      | Equatorward  |
| 2014 (max)               | ~ 47                  | ~ 72                   | 50 – 150             | 30 – 60      | Equatorward  |

# 220 3.3. Solar wind driver and geomagnetic activity dependence of MSTIDs

We have further investigated the MSTID occurrence with respect to different Kp index levels and IMF Bz conditions, shown in Figure 7. We grouped the MSTIDs with respect to Kp; with  $Kp \ge 3$  (geomagnetically active times) and Kp 



the number of MSTIDs in each Kp group to the total number of days within the respective group) is presented in Figure 7a. In both the categories, the occurrence of the MSTIDs was comparatively high during solar maximum years, consistent with their occurrence having a direct dependency on solar activity. In order to check variability of MSTIDs occurrence with respect to IMF Bz conditions, we segregated and normalized the MSTIDs into four groups under prolonged northward, southward, fluctuating, and close to zero IMF Bz. The normalization is done by dividing the number of MSTIDs in a particular IMF Bz category to the total number of days within the respective category for each year. Figure 7b shows the distribution of the normalized MSTIDs occurrence (%) with respect to each category. It is evident that irrespective of the categories of IMF Bz, MSTIDs occurrence was relatively higher during solar maximum than during minimum years (Figure 7b and Table 2). It is interesting to note that MSTIDs occurrence was comparatively higher during prolonged northward and southward IMF Bz conditions (excepting the 1996 minimum). During solar maximum years intervals of steady Bz conditions had more than 80 % MSTID occurrence (100 % during the 2001 maximum). Fisher exact significance test also confirms the results are significant with p value less than 0.05. Small Bz or fluctuating Bz conditions tended to yield relatively fewer MSTIDs.

**Figure 7.** (a) Shows the distribution of the MSTIDs normalized with respect to Kp index. (b) Shows the distribution of the MSTIDs normalized with respect to different IMF Bz conditions.

Table 2. MSTIDs' normalized percentage occurrence with respect to Kp index and IMF Bz.

| Year (solar activity)                       |             | 1996 (min) | 2009 (min) | 2001 (max) | 2014 (max) |
|---------------------------------------------|-------------|------------|------------|------------|------------|
| Normalized<br>Occcurrence<br>wrt Kp (%)     | Kp < 3      | ~ 49       | ~ 51       | ~ 71       | ~ 82       |
|                                             | Kp > 3      | ~ 10       | ~ 30       | ~ 60       | ~ 73       |
| Normalized<br>Occcurrence wrt<br>IMF Bz (%) | Prolonged N | ~ 20       | ~ 77       | ~ 100      | ~ 80       |
|                                             | Prolonged S | ~ 70       | ~ 58       | ~ 86       | ~ 87       |
|                                             | Fluctuating | ~ 40       | ~ 47       | ~ 68       | ~ 72       |
|                                             | Close to 0  | ~ 34       | ~ 49       | ~ 62       | ~ 68       |

We have further checked the variability of the MSTIDs' parameters with respect to different Kp index and IMF Bz conditions. The parameters did not show any variability with different Kp index conditions (figure not provided). Figure 8 shows the MSTIDs' average parameters (velocity, azimuth angle, and period) for different IMF Bz conditions during solar maximum (maroon curve) and minimum (grey curve) years. The parameters did

not show significant variability with respect to solar activity and IMF Bz conditions. There was a relatively higher velocity under steady IMF Bz (Northward and Southward) conditions during solar maximum years (Figure 8a). It is also to be noted that average values of the parameters in each IMF Bz category of solar max year were under 1σ values (shown by error bars) of the solar min year and vice-versa and in the dominant parameters range as shown in Figures 5 & 6 and Table 1.

**Figure 8.** Shows the variability of the MSTIDs' average parameters during solar maximum (maroon curves) and minimum (grey curves) years under different IMF Bz conditions.

## 4. Discussion

In the previous section, we have described the characteristics and occurrence of high-latitude MSTIDs observed over the EISCAT 3D site location and how they vary with different levels of solar activity and with season. We can infer that most of the MSTIDs propagated equatorward (southward) with dominant velocity in the range of  $50 - 150 \text{ m s}^{-1}$  and period in the range of 30 - 60 minutes. Their occurrence showed seasonal variation; they normally occur during winter and equinoctial months. We have also observed that with increasing solar activity the MSTID occurrence increased (Figure 4 & Table 1). In this section we discuss the role of solar forcing in the generation of MSTIDs. In addition, we also compare the results obtained by the MULMEM and FFT method.

## 4.1 Drivers behind the generation of high-latitude MSTIDs

Daytime high-latitude MSTIDs over the EISCAT-3D site were analysed to examine their occurrence and possible generating sources. While previous works have reported their seasonal variations and proposed several generation mechanisms (such as geomagnetic forcing and atmospheric gravity waves), the dominant source remains uncertain. We investigated their occurrence with respect to different seasons and solar activity conditions. Our observations show a clear seasonal variation (Figure 4), with occurrence highest during winter and equinoctial months and lower during summer. This pattern agrees with earlier reports of winter maxima at high latitudes

265










(Ogawa et al., 1987; Negale et al., 2018) and contrasts with summer peaks found in other studies (e.g. Vlasov et al., 2011). The similarity between the present seasonal pattern and that of atmospheric gravity wave activity in the Arctic (Hei et al., 2008; Hoffmann et al., 2010; Yoshiki & Sato, 2000) suggests that gravity waves propagating upward from the lower atmosphere may contribute to MSTIDs generation. However, the transmission of such waves is influenced by stratospheric and mesospheric filtering (Boyde et al., 2025) and possible dissipation at critical levels (Fritts & Vadas, 2008; Vadas, 2007). Confirmation of their role will require dedicated ray-tracing studies to identify specific sources and propagation paths.

In addition to the seasonal variation, MSTIDs occurrence also varies with solar activity (Figure 4c, Table 1), increasing from below 50 % during solar minima to more than 70 % during solar maxima. This behaviour broadly reflects the enhanced geomagnetic activity associated with higher solar flux. Previous studies have disagreed on the extent of geomagnetic control – some linking MSTIDs to Joule heating from storms and substorms (Prikryl et al., 2022, 2024), others noting frequent events during geomagnetically quiet times (Frissell et al., 2016; Ishida et al., 2008). Recently, Moges et al. (2024) investigated MSTIDs amplitude-solar activity dependence and suggested that over high-latitudes the dependency is quite complex involving multiple mechanisms together. Our results show that the occurrence of high-latitude MSTIDs increases with increasing solar activity. We have further investigated their occurrence with respect to different Kp index levels (Figure 7a). However, there is a modest increase in their occurrence during geomagnetic quiet times (Kp 





(RT plot) implies that the time separation of the MSTID bands in this case was less than 60 minutes between 9 and 13 UT with a higher time separation (~90 minutes) between 13 and 15 UT. This is consistent with the period determined using FFT (see blue curve in Figure 9b). The period determined using the MULMEM (see red curve in Figure 9b) was more than 120 minutes. We can infer from these results that MULMEM overestimated the period in this case. This might be due to the approach used by MULMEM, i.e., the automatic selection of the time series, which could result in the selection of time series in which the MSTID structure was not prominent, and hence skipping of some MSTID bands. For the FFT method, on the other hand, we have manually selected the time series where the MSTID bands were prominent in the RT plots to determine the period. To further investigate any disparity in the results, we have compared the average period of all the observed MSTIDs during the four years using both FFT and MULMEM, shown in Figure 10. The red curve shows the average period determined using MULMEM and the blue shows the average period using FFT of each MSTID event. The period determined using both the methods generally shows quite similar values, however there are a few cases during 2009 and 2014 where MULMEM overestimated the period (Figures 10c-d).

These results suggest that MULMEM may have sometimes mischaracterised the parameters of the MSTIDs or, in some cases, failed to identify them. It is worth noting also that the results in this study are based on a single dataset of GBS observations from the HAN radar, which itself showed a seasonal variation. Despite our efforts to account for the variation in GBS it is nevertheless difficult to rule out the possibility that the seasonal variations and mischaracterization of the MSTIDs parameters might have some additional biases incurred due to observational constraints and inefficiency of the analytical method. Therefore, future studies should employ a multi-instrument and multi-method approach to mitigate any such biases in the observations.

**Figure 9.** (a) RT plot shows an example of the MSTIDs. (b) Period of the observed MSTID derived using MULMEM (red curve) and FFT (blue curve).

Figure 10. Shows the comparison of the period of MSTIDs (observed across all four years) derived using MULMEM (red curve) and FFT (blue curve) method.

## 5. Conclusions





This study has presented a statistical overview of high-latitude MSTIDs over the newly established EISCAT-3D radar site. To explore the characteristics and generating sources, ground backscatter data from the Hankasalmi radar during solar maximum (2001 & 2014) and minimum (1996 & 2009) years of solar cycles 23 and 24 have been used. Primary population of MSTIDs propagated equatorward with velocity and period in the range of 50 -150 m s<sup>-1</sup> and 30 – 60 minutes, respectively. Their occurrence showed seasonal (primary peaks during winter and equinoctial months) as well as solar activity dependency (significantly higher occurrence (~72 %) during solar maximum years). Furthermore, their occurrence showed little dependence on Kp index; irrespective of the Kp index conditions, it increased with increasing solar activity. It is interesting to note that MSTIDs showed a comparatively higher occurrence during prolonged northward and southward IMF Bz conditions with more than 80 % during solar maximum years. This increment in the MSTIDs occurrence might be due to the enhanced Joule heating caused by magnetic reconnection during these IMF Bz conditions. It is interesting to note that their parameters (velocity, azimuth, and period) did not show any significant variability with respect to seasons, Kp index, IMF Bz conditions, and solar activity. This study tried to elicit the role of different sources (solar activity, geomagnetic activity, IMF Bz) on the generation of high-latitude MSTIDs but there still exists an uncertainty regarding the dominant drivers. Also, in the determination of the characteristics using the MULMEM method an inconsistency has been observed, which necessitates a future study with multi-instrument and multi-method approach.

Data Availability / Open Data. Data were obtained from the SuperDARN data mirror at the British Antarctic Survey (https://api.bas.ac.uk/superdarn/mirror/v3/). The data were processed utilizing the FITACF 2.5 library


from the Radar Software Toolkit (RST; SuperDARN Data Analysis Working Group, 2025) and pyDARN python

355 library (SuperDARN Data Visualization Working Group, 2025).

Author Contributions. AG and RR conceptualized the idea, performed analyses, and wrote the original manuscript. TKY and ML reviewed and edited the manuscript.

360 *Competing interests.* The authors declare no conflicts of interest relevant to this study.

Acknowledgements. RR and AG acknowledge the financial support from the UKRI NERC funded project NE/W003090/1. TKY is supported by UKRI STFC Grant ST/S000429/1 and NERC grant NE/V000748/1. ML is supported by UKRI STFC Grant ST/S000429/1. The authors acknowledge the use of SuperDARN data. SuperDARN is a collection of radars funded by national scientific funding agencies of Australia, Canada, China, France, Italy, Japan, Norway, South Africa, United Kingdom, and the United States of America. The Hankasalmi radar is maintained and operated by University of Leicester with current support from UKRI NERC under NE/V000748/1.

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
