# Peer review of "High-latitude MSTIDs over the EISCAT-3D site: Solar activity and seasonal dependency"

_EGUsphere, 2025_

## Author Comment (AC1)

**Response to the Reviewer 2**

**Manuscript Number:** egusphere-2025-5769

**Manuscript Title:** High-latitude MSTIDs over the EISCAT-3D site: Solar activity and seasonal dependency

**This manuscript presents a statistical investigation of high-latitude daytime medium-scale traveling ionospheric disturbances (MSTIDs) using ground backscatter observations from the SuperDARN Hankasalmi radar, with a particular focus on the region overlapping the newly established EISCAT_3D site. By comparing solar maximum and minimum years from solar cycles 23 and 24, the authors examine seasonal variability, solar activity dependence, and possible external drivers of MSTIDs. The study is generally well structured and carefully executed. However, several aspects of the interpretation require clarification, particularly concerning observational biases related to ground backscatter, the physical interpretation of the IMF Bz dependence, and the limitations of the analysis methods. For these reasons, this reviewer recommends major revision before being published in Annales Geophysicae.**

We are grateful to the reviewer for their valuable comments/suggestions, which helped us in improving the quality of the manuscript. The responses to the comments are provided below in the same sequence. We have also modified the manuscript based on the reviewer's comments/suggestions.

**Major comments:**

**A key result of the study is the seasonal variation in MSTID occurrence, with higher occurrence during winter and equinoctial months. At the same time, the authors clearly show that ground backscatter (GBS) itself exhibits a strong seasonal dependence. Although the introduction of the "relative occurrence" metric is a reasonable attempt to mitigate this effect, it remains unclear to what extent this approach fully separates true MSTID climatology from observational bias. In particular, it would be helpful if the authors could discuss more explicitly whether the reduced MSTID occurrence during summer reflects a genuine physical reduction in MSTID activity or is primarily a consequence of reduced detectability due to lower GBS occurrence. A clearer discussion of the statistical robustness and limitations of the relative occurrence metric would strengthen the confidence in the seasonal conclusions.**

Response: We appreciate the reviewer's concern about the seasonal dependency of GBS and MSTID occurrence. As mentioned, we have tried to mitigate this with the help of relative occurrence, and we now consider the significance of this between the seasons across all the four years. When paired sample t-test was performed, we found that relative occurrence between summer and winter for all the four years is statistically significant with p-value 0.012.

However, it is worth noting that there are prior works that have studied MSTID climatology using different datasets over both the hemisphere with differing results. Authors like Ogawa et al., 1987 [using NNSS (Navy Navigation Satellite System) satellite data] and Moges et al., 2024a (using ionosonde data) have similarly found MSTIDs' peak occurrence in winter. Whereas, in a contrasting result, Vlasov et al., 2011 found MSTIDs' peak occurrence in summer, although they also mention observational biases. Based on previous and present studies it is evident that single dataset observation will incur some biases, therefore, multi-instrument detailed study is required to have a clear understanding of seasonal variability of high-latitude MSTIDs.

This has also been discussed in the Discussion section of the revised manuscript.

**The dependence of MSTID occurrence on IMF Bz orientation is another important and interesting result of the paper. The finding that MSTIDs are more frequent during prolonged northward or southward IMF Bz conditions is intriguing and consistent with recent studies suggesting a role for reconnection-driven processes. However, the manuscript would benefit from a clearer definition of what constitutes "prolonged" or "steady" IMF Bz, including the thresholds and time intervals used for classification. Furthermore, the physical interpretation deserves more careful treatment, as northward and southward IMF Bz correspond to fundamentally different coupling mechanisms (e.g., lobe reconnection versus dayside reconnection and enhanced Joule heating). While it is plausible that both scenarios can lead to MSTID generation, the manuscript should more explicitly acknowledge these differences and avoid implying a single dominant mechanism without sufficient evidence**

Response: We thank the reviewer for this suggestion.
In order to check variability of MSTIDs occurrence with respect to IMF Bz conditions, we segregated and normalized the MSTIDs into four groups designated as close to zero, prolonged northward, southward, and fluctuating IMF Bz. The segregation is done based on its behaviour in the considered time window of 8-16 UT with the following criteria:
If the temporal average and standard deviation (std) of IMF Bz is within ±1.5nT, it is considered as 'close to zero'. For 'prolonged northward', if the temporal average is greater than 1.5nT with 80% of the IMF Bz values lying northward (> 1.5nT) and remaining northward for at least two hours or more. For 'prolonged southward', if the temporal average is less than -1.5nT with 80% of the IMF Bz values lying southward (<-1.5nT) and remaining southward for at least two hours or more. For 'fluctuating IMF', the temporal average and std of IMF Bz is within ±2nT and more than ±1.5nT respectively with IMF Bz continuously fluctuating between north and south. Figure A shows examples of the four categories of the IMF Bz. The definition of all the terms has been included in the revised manuscript.
Based on the reviewer's suggestion we have carefully acknowledged the role of different IMF Bz configurations and have modified accordingly in the revised manuscript.

[Figure]

**Figure A.** Shows examples of the four categories of IMF Bz.

**The comparison between MULMEM and FFT-derived periods is a valuable component of the study and highlights important methodological limitations. The discussion of cases where MULMEM appears to overestimate the MSTID period is informative, but it would be beneficial to generalize this finding. In particular, the authors could clarify under what conditions MULMEM is most likely to mischaracterize MSTID parameters (for example, weak signals, overlapping wave modes, or automatic time-series selection). Providing a more quantitative assessment of how frequently such discrepancies occur would further enhance the usefulness of this comparison for future studies.**

Response: The authors thank the reviewer for the suggestions. In order to assess the performance of MULMEM in the parameter estimation of MSTIDs, we compared the MSTIDs' parameters obtained from MULMEM with the results from FFT. MULMEM has estimated similar period with that of FFT (with standard deviation < 20% of the mean period) in more than 75% of cases. In rest of the cases MULMEM overestimated/underestimated due to the presence of weak signal or noisy GBS (continuous GBS between the bands). Figure B shows

examples of such cases. In all these cases either GBS signal was very weak (average GBS <= 15 dB) (subfigures a, b, d, e) or noisy (subfigure c). We have also included this aspect in the revised manuscript.

[Figure]

**Figure B.** Shows examples of MSTID cases where MULMEM overestimated the period.

**While the manuscript places the results in the context of previous high-latitude MSTID studies, the definition of "high-latitude MSTIDs" adopted here could be discussed more explicitly. Because the present study focuses on daytime events detected via ground backscatter at specific radar ranges, it is not immediately clear whether the same population of MSTIDs is being sampled as in optical or incoherent scatter radar studies. A short clarification of how the present observations relate to, or differ from, earlier high-latitude MSTID climatologies would help to better position the contribution of this work.**

Response: The authors appreciate the suggestion given by the reviewer. We have duly added the definition of high-latitude MSTIDs in the revised manuscript: MSTIDs observed between

60º and 90º latitude are categorized as high-latitude MSTIDs. As the HAN radar probes the same ionospheric region (as shown by its FOV in Figure 1b in the manuscript), the main objective of this study is to investigate the propagation characteristics and occurrence of high-latitude MSTIDs during different solar activity conditions.

We have also added clarification related to the present observation in the discussion section of the revised manuscript: We investigated their occurrence with respect to different seasons and solar activity conditions. Our observations show a clear seasonal variation (Figure 4 in the manuscript) with a dependency on solar activity. Their occurrence is highest during winter and equinoctial months and lower during summer. Previous studies investigated MSTID occurrence utilizing datasets from different instruments such as ionosondes, incoherent scatter radars, NNSS satellites, etc. Some studies have reported their characteristics, others have mentioned the complex dependency of their amplitude on solar activity, and a few have reported their seasonal variations (Moges et al., 2024a & 2024b; Negale et al., 2018; Ogawa et al., 1987; Vlasov et al., 2011). The present study encompasses not only the characteristics and seasonal variation but also reports their dependency on IMF Bz configurations and solar activity. Overall, our findings indicate that both internal atmospheric processes and external solar forcing contribute to MSTIDs generation.

**Minor comments:**

**The abstract and conclusions repeatedly emphasize the relevance of the results for future EISCAT_3D operations. While this is an important motivation, some repetition could be reduced to improve conciseness**

Response: We have removed a number of repeated instances of mentioning the EISCAT 3D site. We note that it is only mentioned once in the conclusions, and so this instance has been left in.

**Line 112: How do the authors select the offsets between the three cells, i.e., plus-minus 2 in beam and 4 in range. If you change these offsets, would you have different results?**

Response: We thank the reviewer for raising this question. In the present study the cell set combination with separation of ±2 in beam and 4 in range gate is considered to maintain a quasi-constant equilateral separation between the three cells. As MSTIDs typically have wavelengths of a few hundred kilometres, the separation in range is 4 cells, corresponding to 180 km at the ground (45 km coverage per range gate) and ~100 km at the ionospheric reflection point. There can also be other cell set combinations which maintain the quasi-constant equilateral separation (mentioned in Grocott et al., 2013). But the rationale behind using only the aforementioned cell set combination is that MULMEM is computationally

exhausting and executing on a large dataset spanning over four years with multiple MSTID events becomes challenging. However, we have tested a few events with different cell set combinations (beam separation ±2 and the range separation of 2) and compared the results obtained of one of the MSTID events with the previous cell set combination (as shown in Figure C). Figure D presents the cell-wise average of the parameters as shown with red ([(5, r); (7, r+4); (9, r)] and green ([(5, r); (7, r+2); (9, r)]) markers. It can be seen from Figures C and D we get similar results with both cell set combinations.

[Figure]

**Figure C.** Shows temporal variations of the parameters obtained (a) using cell set combination of [(5,r); (7,r+4); (9,r)] and (b) using cell set combination of [(5,r); (7,r+2); (9,r)], where (5,7,9) are the beam numbers and 'r' is the range gate.

[Figure]

**Figure D.** Shows the average parameters obtained using the two cell set combinations (as shown in Figure C).

**Figure 2: Meaning of black rectangles should also be described in the caption**

Response: The suggested modification has been incorporated in the caption of Figure 2 in the revised manuscript.

**Line 117-118: The authors describe that the derived parameters are used for statistics if they are similar between different cell sets. What is the procedure of this data selection?**

Response: We thank the reviewer for pointing this aspect. In order to check the similar values between different cell sets, we determined the standard deviation of the parameters obtained from different cell sets. If the standard deviation lies within 20% of the mean value, we have considered the parameters to be similar. We have also added this aspect in the revised manuscript.

**Figure 4d: What is the unit of the relative MSTID occurrence?**

Response: To determine the relative occurrence, we have divided the MSTIDs percentage occurrence (i.e., number of MSTIDs per month) for each month with the percentage occurrence of GBS (number of GBS per month) for the respective month. So, the relative occurrence is a parameter without unit.

**Figure 6: Unit (m/s) should be added to the labels in the radial axis**

Response: The suggested modification has been incorporated in Figure 6 in the revised manuscript.

**Line 210: This reviewer recommends the authors describe how the significant test was applied to the results in more detail**

Response: We have performed the significance test (Fisher Exact test) for the obtained results. Fisher Exact test is a statistical procedure to determine the probabilities (p value) of categorial variables (Fisher, 1935), which works with the initial consideration that categories are independent. In the present study, we have used this test between two categories (meridionally and zonally propagating MSTIDs) and we found that differing velocities of zonally and meridionally propagating MSTIDs is significant (p value < 0.0001). The suggestion has also been incorporated in the revised manuscript.

**The criteria used to classify IMF Bz conditions (e.g., thresholds and time windows) would be clearer if summarized in a table or an appendix. In particular, how do the authors evaluate the fluctuation level of IMF Bz?**

Response: We have described the criteria to classify IMF Bz conditions in the response to major comment 2 and also in the revised manuscript.

**There appears to be a typographical error in the discussion section where "MMF Bz" is used instead of "IMF Bz"**

Response: The typographical error has been corrected in the revised manuscript.

**Although the overall quality of the English is good, several long sentences in the Discussion section could be shortened to improve readability**

Response: Based on the reviewer's suggestion, the authors have tried to modify and shorten few of the sentences in the Discussion section in the revised manuscript.

**References:**

Grocott, A., K. Hosokawa, T. Ishida, M. Lester, S. E. Milan, M. P. Freeman, N. Sato, and A. S. Yukimatu: Characteristics of medium-scale traveling ionospheric disturbances observed near the Antarctic Peninsula by HF radar, Journal of Geophysical Research: Space Physics, 118, 5830–5841, https://doi.org/10.1002/jgra.50515, 2013.

Fisher, R. A. The logic of inductive inference. Journal of the Royal Statistical Society, 98, pp. 39-82, https://doi.org/10.2307/2342435, 1935.

Moges, S. T., Sherstyukov, R. O., Kozlovsky, A., Ulich, T., & Lester, M.: Statistics of traveling ionospheric disturbances at high latitudes using a rapid- run ionosonde, Journal of Geophysical Research: Space Physics, 129, e2023JA031694, https://doi.org/10.1029/ 2023JA031694, 2024a.

Moges, S. T., Kozlovsky, A., Sherstyukov, R. O., & Ulich, T.: Solar activity dependence of traveling Ionospheric disturbance amplitudes using a rapid-run ionosonde in high latitudes, Journal of Geophysical Research: Space Physics, 129, e2024JA033013, https://doi.org/10.1029/2024JA033013, 2024b.

Negale, M. R., Taylor, M. J., Nicolls, M., Vadas, S. L., Nielsen, K., & Heinselman, C. J.: Seasonal propagation characteristics of MSTIDs observed at high latitudes over central Alaska using the poker flat incoherent scatter radar, Journal of Geophysical Research: Space Physics, 123(7), 5717–5737, https://doi.org/10.1029/2017ja024876, 2018.

Ogawa, T., Igarashi, K., Aikyo, K., & Maeno, H.: NNSS Satellite-observations of medium-scale traveling ionospheric disturbances at southern high-latitudes, Journal of Geomagnetism and Geoelectricity, 39(12), 709–721, https://doi.org/10.5636/jgg.39.709, 1987.

Vlasov, A., Kauristie, K., van de Kamp, M., Luntama, J.-P., and Pogoreltsev, A.: A study of Traveling Ionospheric Disturbances and Atmospheric Gravity Waves using EISCAT Svalbard Radar IPY-data, Annales Geophysicae, 29, 2101–2116, https://doi.org/10.5194/angeo-29-2101-2011, 2011.